# Outcome Disparities in Patients with Early-Stage Laryngeal Cancer Depending on Localization, Tobacco Consumption, and Treatment Modality

**DOI:** 10.3390/biomedicines12092136

**Published:** 2024-09-20

**Authors:** Theresa Wald, Tim-Jonathan Koppe, Markus Pirlich, Veit Zebralla, Viktor Kunz, Andreas Dietz, Matthaeus Stoehr, Gunnar Wichmann

**Affiliations:** 1Clinic for Otorhinolaryngology, Head and Neck Surgery, University Hospital Leipzig, Liebigstr. 10–14, 04103 Leipzig, Germany; tim-jonathan.koppe@medizin.uni-leipzig.de (T.-J.K.); markus.pirlich@medizin.uni-leipzig.de (M.P.); veit.zebralla@medizin.uni-leipzig.de (V.Z.); viktor.kunz@medizin.uni-leipzig.de (V.K.); andreas.dietz@medizin.uni-leipzig.de (A.D.); matthaeus.stoehr@medizin.uni-leipzig.de (M.S.); gunnar.wichmann@medizin.uni-leipzig.de (G.W.); 2The Comprehensive Cancer Center Central Germany, Leipzig University Hospital, Liebigstr. 10–14, 04103 Leipzig, Germany

**Keywords:** head and neck squamous cell carcinoma (HNSCC), glottic cancer, supraglottic cancer, outcome research, early stage, smoking, therapy, TNM staging

## Abstract

**Background/Objectives:** Laryngeal squamous cell carcinoma (LSCC) is among most frequent malignancies of the head and neck. Recent oncologic research focusses on advanced rather than on early stages. Thus, we aimed to improve the knowledge concerning prognostic factors and survival in early glottic (GC) and supraglottic cancer (SGC). **Methods:** We retrospectively investigated patients diagnosed in 2007 to 2020 with stage I or II GC (ICD-10-C32.0) or SGC (ICD-10-C32.1, C32.8 or C32.9). For precise discrimination of GC and SGC, pathology reports about biopsy and definitive excision were closely examined and information on clinical characteristics and risk factors were collected before analyzing patterns of risk factors for overall survival (OS) in multivariate Cox regression analyses (mvCox). **Results:** The cohort included 220 patients with early GC (*n* = 183) and SGC (*n* = 37). The GC patients showed significantly improved 5-year OS compared to SGC patients (83.6% vs. 64.9%; *p* = 0.004), whereas survival according to UICC stage (I vs. II) was not different (*p* = 0.177). Surgical resection was superior to definitive radiotherapy (RT) for 5-year OS (*p* < 0.001). Cumulative tobacco consumption of greater than 10 pack years drastically impaired OS (*p* = 0.024), especially in patients receiving RT (*p* < 0.001). Supraglottic localization, smoking, and re-resection after initial R1 status consistently were independent prognostic factors in mvCox. **Conclusions:** Our cohort of early LSCC patients demonstrates significant negative impact of supraglottic localization, older age, tobacco consumption, poor tumor differentiation, and re-resection on OS. Further research is required as there is still lack of evidence on optimal decision-making and therapeutic strategies.

## 1. Background

The majority of malignancies in the head and neck area originate from the squamous epithelium of the upper airways and are summarized under the term head and neck squamous cell carcinoma (HNSCC) [1]. In Germany, laryngeal cancer was responsible for around 1% of all new cancer cases in men and 0.4% in women in 2019/2020, ranking it the second most common localization of HNSCC after cancer of the oral cavity [2]. Additionally, global cases of laryngeal cancer increased during the last decade, with around 209,000 new cases in 2019, i.e., a plus of 24.7% compared to 2010 [3]. However, laryngeal squamous cell carcinoma (LSCC) still remains a rare disease. The main risk factors for developing LSCC are tobacco smoking—especially when coinciding with higher level alcohol consumption—low socioeconomic status, male sex, and exposure to certain environmental and occupational hazards such as asbestos, cement dust, and other particles [4,5,6].

Generally, about half of all patients presenting with laryngeal cancer find themselves with an advanced stage neoplasia at time of initial diagnosis, drastically impairing the outcome compared to early-stage disease [7].

When determining the treatment modality for patients, the main deciding factor is the stage of the cancer based on the current UICC- and TNM-classification at the time of diagnosis. For early stages (UICC I/II, T1/T2 N0 M0), evidence-based therapeutic options are monomodal surgery or radiotherapy, which claim to be equivalent in achieving favorable survival and preserving post-therapeutic larynx function such as breathing, speaking, and swallowing. For advanced stages (UICC III/IV), treatment protocols contain a multimodal approach combining surgery, irradiation, and chemotherapy, for instance, concurrent chemo-radiotherapy (CRT) and upfront surgery followed by risk-adapted post-operative (adjuvant) radiotherapy without (PORT) or with concurrent platinum-based chemotherapy (PORCT), whereas definitive radiotherapy should only be considered in case of impaired general health and limited organ function not allowing simultaneous chemotherapy [8]. Multimodal regimens are deemed to be inadequate for treatment of early stages, as the good oncologic outcome regarding survival is paid by highly increased acute and late toxicity.

Most cases with laryngeal cancer of the glottis first present with hoarseness [9]. This is mostly discussed as the main reason behind the observation that the glottic localization of the cancer correlates with a significantly better outcome for the patients compared to subglottic or supraglottic involvement [10]. On the one hand, this is due to the early development of the symptom hoarseness and thus, presentation in an early stage at the first doctor’s visit, and on the other hand due to the rare lymphatic drainage of the glottis, preventing early locoregional metastasis. Involvement of more than one part of the glottis and/or the supraglottis is linked to impaired outcome; e.g., Tulli et al. showed that involvement of both the supraglottic area and the anterior third of the anterior commissure corresponds to a worse outcome [11].

As the extent of alcohol and tobacco consumption are main risk factors for development of HNSCC including LSCC, it is reasonable that adhering to these lifestyle-related risk factors was shown to exert a negative impact on the patient’s survival. Similarly, certain tumor specific characteristics such as grading are also claimed to be important. The HPV-association was not found to be an elemental risk factor of larynx cancer, unless the underlying disease is laryngeal papillomatosis, which is rare [12].

Research often focuses on local and locoregional advanced (LA) tumor stages, probably because of the much higher headroom for improvement in LA LSCC and the lower- case number of patients required to demonstrate significant outcome differences between groups, but also because of the generally good outcome of patients with early glottic (GC) or supraglottic cancer (SGC). However, there is a lack of information about various prognostic factors, survival, and functional outcome considering the available treatment options. Studies on early GC and SGC often have a retrospective design and report outcomes of small numbers of patients. Underlining this, in the last Cochrane review written in 2014 [13] examining the outcome after surgery vs. radiation in early GC patients, only one trial could be included, which was already published in 1990 and has a high risk for systematic bias [14].

Therefore, we were interested in long-term follow-up data of patients with early GC and SGC treated at a comprehensive cancer center to analyze risk factors to facilitate therapeutic decision making.

## 2. Methods

The tumor database of the department of otolaryngology of the University Hospital Leipzig includes a total of 7134 patients. We extracted data of 1285 patients with the larynx as primary tumor site, diagnosed from 1993 to 2020. The database lock for this retrospective study was 15 November 2023. The selection process and criteria for eligibility are pictured as CONSORT diagram in Figure 1. As the multidisciplinary tumor board (MDTB) with pre-therapeutic decision making was established at our certified cancer center in 2007 and utilized comprehensive presentation of the case including standardized imaging, we focused on the subsample of patients with identical diagnostics and decision-making processes and presentation to the MDTB after treatment, allowing reliable recording of the patients’ path including diagnostics, treatment, and follow-up.

Main inclusion criteria were: (1) Date of initial diagnosis from 2007 onwards (date of initial diagnosis was elected to be the date of histopathological confirmation of squamous cell carcinoma (SCC) after biopsy); (2) Pathologically verified SCC (International Classification of Diseases for Oncology, third edition ICD-O-3 8070/2, 8070/3, 8071/3); (3) Pathologically verified, early-stage glottic or supraglottic laryngeal cancer (Tis, T1, T1a, T1b or T2, without locoregional or distance metastasis N0 M0 stage I or II according to TNM 2010 classification, and localization and extent of disease coded as C32.0, C32.1, C32.2 or C32.8, C32.9 according to the 10^th^ version of the International Statistical Classification of Diseases and Related Health Problems, ICD-10); (4) Patients without a synchronous or meta-synchronous secondary tumor.

Statistical analyses using SPSS version 29 (IBM Corporation) included *Pearson* chi-squared tests to assess differences between categorical variables and *Student’s t* tests for numerical covariates. Outcome differences between groups were analyzed using Kaplan-Meier cumulative survival plots and log-rank tests. Univariate and multivariate *Cox* proportional hazard regression analyses were used to estimate a covariate’s hazard ratio (*HR*) and to identify independent predictors of outcome parameters. All *p* values < 0.05 in 2-sided tests were considered significant.

Time-dependent outcome parameters, i.e., overall survival (OS), tumor-specific survival (TSS), event-free survival (EFS), and time to non-cancer related death (NCRD), were calculated from the date of initial diagnosis to the corresponding date of an event. Furthermore, parameters concerning relapse of disease were analyzed, i.e., locoregional relapse-free survival (LRRFS); time to event regarding diagnosis of locoregional relapse, also known as loss of locoregional control (LRC); local control (LC; LRFS), nodal control (NC; NRFS), distant control (DC; DMFS), and occurrence of other cancer.

## 3. Results

We collected 225 patients with clinically early GC and SGC meeting the inclusion criteria, of whom 184 patients (81.8%) were diagnosed with GC and 41 patients (18.2%) with SGC. After tumor resection including guideline-based neck dissection and histopathological examination, 5 patients with clinical stage I (*n* = 1, SGC) and II (*n* = 4, 1 GC and 3 SGC) were reclassified (upstaged) to higher UICC stages due to nodal involvement, with 2 patients ending up with UICC stage III tumors (pT1pN1 and pT2pN1), while the remaining 3 patients were UICC stage IVA (pT2pN2b)—resulting in their exclusion from further analyses. The final cohort consists of *n* = 220 (GC *n* = 183 and SGC *n* = 37). 18 patients were found to have *carcinoma in situ* and were classified as UICC stage 0. The characteristics of the cohort are shown in Table 1. 92.3% (203/220) of all patients were male. The mean age was 64.7 (95% CI 63.4–65.0; median 64.4, range 44.2–87.3) years. Patients with SGC were slightly younger and more likely to be female than those with GC. The SGC patients reported higher exposure to lifestyle risk factors according to pack years of tobacco smoking history and higher consumption of alcohol in higher amounts per day. They included a higher proportion of T2-cases compared to patients with GC, while not including even one patient with *carcinoma in situ*.

The predominant treatment was surgery, as 209 patients were primarily treated by surgery and 11 patients received primary radiotherapy. The 209 primary resected include a total of 82 patients with unilateral or bilateral neck dissections. Only 2 of the patients ending up with R1-resection status were not amenable for larynx organ-preservation partial resection and received full-dose adjuvant radiotherapy. One patient with pT2 pN0 M0 SGC received adjuvant combined radiochemotherapy for an unknown reason, possibly because of a close R0-excision margin of 2 mm. Two other patients received adjuvant radiotherapy for R1-resection, 1 patient with pTis GC and 1 patient with 2 concurrent pT1a GC of the anterior commissure and the posterior third. Due to larger lesions and according to the decision of the MDTB, 6 patients with UICC stage II GC (*n* = 3) and SGC (*n* = 3) were referred to adjuvant radiotherapy after tumor resection.

The mean follow-up (FU) in our outpatient clinic was 64.8 months (median FU 59.5 months; range 1.7–178.2 months). The 5-year OS was 81.8% in GC patients and 64.9% in SGC patients (*p* = 0.004) with all patients having N0 and T1/T2-tumors. The TSS was 95.2% for GC and 91.9% for SGC patients (*p* = 0.234). In total, 67 deaths occurred with 45 deaths from other causes (NCRD), 20 HNSCC-associated deaths (tumor-specific death, TSD) and 2 deaths from other cancers. Within 5-years FU, 43 deaths occurred (31 NCRD, 11 TSD, 1 death from other cancer).

### 3.1. Localization

Thirty-seven patients with cancer of the supraglottic space were included. They were confirmed as having supraglottic cancer without coding errors in the ICD-10 classification system. Of the 37, 33 had SGC without glottis involvement. Among the remaining 183 GC patients, 21 patients (26.4%) had cancer originating from the glottis with minor involvement of a supraglottic structure. Patients with glottic carcinoma and those with minor supraglottic involvement showed no difference in survival (*p* > 0.348).

The detailed localization of the tumor differed among the subregions of the glottis: the anterior third was involved in 56.4%, the middle third in 48.2%, and the posterior third in 20.5% (*n* = 124, *n* = 106, and *n* = 45 of 220 patients, respectively). The anterior commissure was involved in 22.7%, the posterior commissure or the region between the arytenoids in 0.5%. A total of 42.7% of the cancers involved more than one area. The OS and TSS did not differ between the localization of the cancer according to the subregions.

OS according to tumor location and UICC stage differed only with respect to location, but not within UICC stage I or II (*p* = 0.177, Figure 2A). Patients with UICC stage I and II GC had better survival than patients with UICC stage I and II SGC (Figure 2B). There were no significant differences in TSS regarding UICC and localization (*p* = 0.693).

However, 5-year OS of GC (ICD10-C32.0) and SGC (ICD10-C32.1) patients differed significantly (*p* = 0.004, Figure 3A), possibly due to significant differences in patient characteristics according to tumor localization. Patients with SGC were more likely to be heavy smokers (>10 pack years, PY) and to have a G3 grading category. The 5-year OS and TSS for T1a and T1b glottic cancer were not statistically different (*p* = 0.434 and *p* = 0.288). Approximately two thirds of SGC patients had UICC stage II cancer, while only one quarter of GC patients had UICC stage II cancer. 

In patients with SGC, grading was not linked to different OS (*p* = 0.956). On the other hand, among GC patients, those with G1/G2 grading had significantly improved survival compared to those with G3 GC (*p* < 0.001). In addition, when comparing the survival among patients with G1/G2, the patients with G1/G2 GC showed a superior survival compared with G1/G2 SGC patients (*p* = 0.001). As expected, the OS for patients with SIN III neoplasia was best, with no death recorded.

### 3.2. Therapy

Primary treatment, i.e., definitive surgery or radiotherapy, did not influence the 5-year TSS (*p* = 0.496). However, the 5-year OS was drastically impaired in patients receiving definitive radiotherapy compared to surgery ± postoperative radio (chemo) therapy (*p* < 0.001; Figure 3B). 72.7% (8/11) of patients treated with definitive radiotherapy had UICC II stage disease. However, 11.3% (8/71) of UICC stage II patients were treated primarily with surgery. Patients with UICC stage II cancer had significantly worse OS (*p* = 0.003) and NCRD (*p* < 0.001), but not TSS (*p* = 0.564), when treated with primary radiotherapy. Whereas survival for patients with smoking history ≤ 10 PY did not differ between the two primary treatment options (*p* = 0.618), smoking > 10 PY was associated with strongly reduced survival after primary radiotherapy (*p* < 0.001), indicating that surgery was best for patients with >10 PY.

Patients without definitive surgical approach or re-resection after initial R1-resection had impaired 5-year OS compared to patients with R0-resection at initial surgery (*p* = 0.001, Figure 3D). This effect translates to the subgroup of patients with a cumulative tobacco consumption of >10 PY who underwent re-resection or definitive radiotherapy (*p* < 0.001, Figure 3F). There was no difference in survival among patients smoking ≤ 10 PY with initial R0 resection, re-resection after initial R1 resection, or definitive radiotherapy (Figure 3E, all *p* > 0.253). In general, the 5-year OS and NCRD were impaired in smokers with a cumulative exposure to >10 PY (*p* = 0.024 (Figure 3C) and *p* = 0.077, respectively), with smoking being an independent predictor of rather poor OS and NCRD. The same was found for re-resection as both analyses, Kaplan-Meier plots and multivariate *Cox* proportional hazard regression models, demonstrated a link between re-resection and reduced OS and NCRD. However, even more detrimental was definitive radiotherapy as the primary treatment (Figure 3D), but this effect was mainly driven by worse OS and NCRD of smokers with >10 PY (Figure 3F), whereas nonsmokers and smokers ≤ 10 PY were not at increased risk when primarily treated by radiotherapy (Figure 3E).

Detailed analyses of predictors using multivariate *Cox* proportional hazard regression models are shown in Figure 4 and demonstrate inferior outcome of early larynx cancer with primary localized in the supraglottis for OS, TSS, NCRD and EFS. Dependent on low numbers of tumor-specific deaths, only three independent predictors (*Pi*) could be elucidated: smoking > 10 PY, re-resection, and localization. The competing risk for non-cancer-related death (NCRD) exerted the main impact on OS, and with slightly volatile exact numbers for hazard ratios, the *Pi* were the same. Because of the low frequency of events, *Cox* proportional hazard regression models analyzing predictive factors for local, nodal, locoregional, and distant control, and progression-free survival could not be calculated.

## 4. Discussion

Based on our monocentric retrospective analysis with complete tumor board documents and pathology reports, we demonstrated a significant difference in survival of 220 patients with early larynx cancer according to the localization of the primary in the glottis or supraglottis, exceeding the survival differences between UICC stage I and II patients according to TNM categories. Whereas staging failed to demonstrate differences in various survival measures including OS, TSS, NCRD, and EFS, the impact of successful R0 resection with R0 ≥ 1 mm for GC and R0 ≥ 5 mm for SGC, a rather low frequency of re-resection requirement after initial R1 resection (14.6% vs. 8.8% based on 85.4% R0 ≥ 1 mm in GC vs. 91.2% R0 ≥ 5 mm in SGC; *p* = 0.581), and even more the avoidance of definitive radiotherapy for the primary tumor was noticed. As there was a strong link between smoking > 10 PY and impaired OS after definitive radiotherapy, mainly resulting from higher numbers of NCRD, caution might be advised when choosing a non-surgical approach for treatment of smokers.

Reviewing literature, the 5-year OS for GC ranges between 82–100% for UICC stage I and 88.8–97.4% for UICC stage II tumors [8,13,14,15]. The 5-year TSS is similarly good. Our data supports these survival rates for GC (5-year OS was 83.6% and the 5-year TSS was 95.6%). However, there was a significant difference to the OS of SGC patients with 64.9% (the 5-year TSS was 91.9%). This might be seen as mainly caused by an unequal distribution of risk factors concerning tumor stage, grading, smoking behavior, and sex. However, our multivariate analysis consistently revealed improved survival of GC patients. Multivariate Cox proportional hazard regression models confirmed localization, grading, and smoking behavior as independent predictors of OS but not of tumor stage. In this regard, it may be questionable, if a sharp cut regarding stage can be made according to the highest diameter of the lesion and omitting other relevant information, such as organ function (e.g., subclinical infiltration of the arytenoid just before causing fixation, but influencing regular vibration of vocal folds and false vocal folds already impairing voice quality), smoking, grading, and other risk factors.

Even in univariate analyses, staging according to UICC fails to predict survival, as there was no discrimination between stage I and II. This is due to GC patients with UICC I and II showing better survival rates than those with SGC and UICC stage I and II. The TNM classification and staging according to UICC seems to be suitable to describe the relevance of the extension of a glottic cancer regarding OS. However, there were no differences between T1 and T2 category in oncologic regards. An even more important factor was the localization of the tumor (glottic vs. supraglottic). Patients with SGC showed impaired OS and NCRD compared to GC patients, regardless of tumor extension. This could be a result from additional lifestyle-related risk factors at play in SGC, for instance alcohol consumption that hardly could be made responsible for GC. However, the number and events within our rather small SGC sample did not allow for a reliable analysis of this factor.

Patients receiving re-resection or radiotherapy had decreased OS (and NCRD) compared to those with initial R0 resection when they had an accumulated smoking history of >10 PY. Smokers with >10 PY were more likely to receive a re-resection. Moreover, radiotherapy was inferior to surgery. Surgery and RT showed equal survival data only within patients with ≤10 PY and non-cancer-related death, leading to the thesis that decision making for therapeutic procedure disregarding smoking status is potentially harmful.

Many studies report on the outcome of patients with early laryngeal cancer, most of them utilizing retrospective data acquired from a single institution. However, in the latest Cochrane review on this topic, the authors Warner et al. [13] identified only one randomized controlled trial (RCT), conducted in the 1980s, that compared open surgery and radiotherapy in patients with T1 and T2 glottic cancer. The potential of systematic bias in this study was estimated as high [14]. No sufficient data for transoral laser surgery was found. A few years later, the same authors [10] reported statistically significant differences and increased OS and TSS favoring transoral endoscopic surgery in a systematic review. Voice quality was not different according to treatment modality, although objective phoniatric parameters were improved in patients who had received radiotherapy. Other reviews report improved oncological outcome after surgery over radiotherapy [16,17]. Again, for T2 GC tumors, Warner et al. [18] reported no difference in local control after RT or surgery.

A prospective EaStER trial failed because of low numbers accrued. Only 15 patients were randomized from 2005 to 2008 in 5 study centers within the UK. Possible reasons for low recruitment might be the study design, and recruiters’ and patients‘ preferences regarding treatment modality [19].

The EORTC 1420 “Best of“ phase III trial aim to assess the “best of” radiotherapy *versus* transoral surgery for patients with T1-T2, N0-N1 oropharyngeal and supraglottic carcinoma, and T1N0 hypopharyngeal carcinoma, hopefully providing recommendation for best treatment for early supraglottic cancer [20]. However, it remains unclear whether the scheduled case numbers will allow for reliable comparison of the multitude of subgroups with/without locoregional metastasis and treatment modalities, due to probably limited power for comparing stage I and II SGC.

In our cohort, when aiming to elucidate the potential impact of sub-localization and substructures of the glottis involved, there was no obvious effect. We found no outcome differences attributable to more detailed (sub-) localization of the tumor within the glottis or caused by involving more than one substructure. However, some studies report a negative influence on survival of involvement of the anterior commissure [11,21], due to the proximity to the thyroid cartilage and its lymphatic vessels and—because of the high risk for developing a synechy—a rather impairing side effect regarding functional outcome, particularly in T1b tumors (involvement of the anterior commissure). However, to the best of our knowledge, a strong impact on survival was not reported.

High-level smoking is linked to increased mutational burden and is carcinogenic, whereas grading reflects increasingly higher levels of de-differentiation and loss of organ-specific gene expression control. Consistently, smoking above a certain level (>10 PY in our analyses) and G3 are widely known as negative prognosticators for survival. However, there seems to be a lack of evidence concerning the strength of effects in early GC and SGC, as most studies include both early- and late-stage cancer. Consistent with our findings, Mucha-Malecka et al. [21] report a significantly worse local control rate and disease-specific survival in smokers with T1 GC undergoing radiotherapy. In their study, grading had no significant influence on survival. To the contrary, our data suggests a cumulative smoking dose of >10 PY and a grading of G3 are independent negative predictors of survival. Patients with G3 tumors had a 2.4-fold increased risk of death from any cause, and also a decreased EFS and NCRD (Figure 4). Smoking was found to be an independent predictor of OS, TSS, NCRD, and EFS, and patients who accumulated >10 PY had a 3.2-fold increased risk of death from any cause. The higher frequency of these risk factors in SGC patients might be associated with the impaired survival compared to GC patients. Jointly, they may increase their detrimental impact on survival when present in combination with other factors, including age.

The negative impact of tobacco smoking on carcinogenesis and the prognosis if continued during or after tumor therapy is well known [7,8,22]. Smoking is associated with particular mutation signatures and seems to predominantly induce genetic alterations damaging the regular course of the cell cycle, causing mutations and changes in gene expression patterns including the down-regulation of tumor suppressor proteins and the enhancement of oncogenic proteins [23].

Another recent study [24] exploring the link between inflammatory biomarkers and head and neck cancer highlights that smoking not only amplifies the inflammatory response, but also contributes to creating a milieu conducive to cancer development and progression. Smoking is associated with elevated levels of M-CSF and the activation of pathways that promote immunomodulation and angiogenesis, indicating a more aggressive cancer phenotype. Smoking is also linked to increased levels of pro-inflammatory cytokines IL-6, IL-8, as well as chemokines such as monocyte chemoattractant protein 1 (MCP-1, CCL2), and growth factors such as vascular endothelial growth factor (VEGF) [25]. In contrast, non-smoking HNC patients display higher levels of immune-enhancing cytokines like IL-10 and IL-15, which might suggest a more effective antitumor response [24]. These findings in context of recent papers by Suzuki et al. [26] and Schoetz et al. [27], highlighting the relevance of through-irradiation-induced, pro-inflammatory, and senescence-associated cytokines (like IL-6) and their role in radioresistance of HNSCC, enhance our comprehension of the pathophysiological mechanisms linking smoking and inferior outcomes of smokers when their tumor was irradiated.

No patient with subglottic cancer was found in our cohort. Several analyses found a prevalence of 1% to 2% for subglottic cancer among larynx cancer patients [28,29]. Accordingly, we had to expect about 3 (3.3, 95% CI 2.00–4.67) patients in our cohort to have subglottic cancer. This difference might be due to our search for early cancer stage with T1/T2 N0 tumors, whereas the vast majority of patients (>80%) with subglottic cancer presented in advanced stages at time of diagnosis. However, as subglottic cancer does not share the main characteristics of early GC and SGC, our analyses are not biased in an inappropriate way.

We had to exclude 5 patients with clinical stage cN0 who were reclassified as pN+ after guideline-based selective neck dissection (SND). Of these, 4 patients had SGC, with one staged as cT1cN0 and three staged as cT2cN0, as well as one GC patient with cT2N0. Elective neck dissection was performed in SGC and T2 GC, according to the German national guideline for surgical resected larynx cancer [8]. This approach revealed an advanced tumor stage with occult locoregional metastasis in 4/41 = 9.8% of SGC and 1/184 = 0.5% of GC. This demonstrates the potential underestimation of the cancer stage by using clinical staging methods only, especially in SGC. Based on the about twentyfold increased risk of missing occult neck metastases when omitting SND, we recommend SND as an essential surgical approach leading to histopathologic tissue examination of the removed lymph nodes, favoring adequate stage-dependent treatment of the patient with SGC.

One strength of our study is the number of patients included, as there are only a few studies on this topic exceeding *n* = 50 patients, and in most cases they also lack sufficient follow-up time. Using ROC curves, we identified the optimal evaluation period for overall survival within a follow-up time of 48 months. In multivariate Cox proportional hazard regression analyses, the models achieved the highest Chi-square when calculated for right-censored follow-up data at 48 months. After 48 months, the treatment-related and tumor-specific effects especially diminish, and artefact susceptibility for many biases increases, as random effects time-dependently increase variance. Because a follow-up period of 60 months is commonly used, analyses for 60 months (although exceeding 48 months) are shown in this publication. In total, events were quite rare and deaths from other causes had a huge impact on OS, underlining the comparably good outcomes of early-stage GC and SGC. However, an older patient population might be even more affected by treatment-related side effects, for instance through radiation, thereby increasing the vulnerability of the patients independent of tumor characteristics as well as increasing NCRD, when the patients are smokers (see above).

Within our study, we confirmed a survival rate of our patients as it would have been expected for (not too old) patients with early GC and SGC. However, survival did not differ between UICC stages I and II. Thus, the staging system can be used to describe the extension of the tumor, but is unable to provide a sufficient prognosticator for reliable estimation or prognostication of survival. More relevant prognostic factors are localization (glottic vs. supraglottic), smoking, and grading, and these factors must thus be obeyed in therapeutic decision making, especially as smoking > 10 PY is associated with impaired survival when receiving definitive radiotherapy instead of primary surgery.

Our study suffers the common limitations of studies in early GC and SGC regarding case numbers in subgroups (T1/T2N0 SGC in particular) and the number of events detected in early cancer including GC and SGC in general. By presenting data from a single tertiary clinic and certified head and neck cancer center, transferability of our findings to any other clinic’s outcome results could be limited. However, in an optimized environment with close collaboration of specialists among all disciplines involved in the treatment of head and cancer patients, including early GC and SGC, a positively biased outcome could be obtained, and it is possible that differences in outcome after particular treatment modalities can only be elucidated in such environment. However, such a positive bias in outcome would be welcome for every patient treated.

More clinical studies are needed, as many publications in this topic date back to before year 2000 and/or analyzed small patient cohorts. As such, an RCT will be very time-consuming and require a (much too) large sample of patients available only in multicenter, at best multi-national trials, to have a sufficiently large number of events for reliable statistics. The costs for such RCT are expected to be prohibitive. Therefore, we ask for setting up a multicenter-registry study to obtain a sample large enough to address all the remaining open questions and gain reliable real-world evidence.

## 5. Conclusions

Smoking > 10 PY, poor differentiation (G3) of tumor tissue, age, and re-resection for achieving R0-status after initial R1-resection are negative prognostic predictors for survival in (mostly elderly) patients with early GC and SGC. The negative impact of these factors on survival was even stronger in patients treated with radiotherapy. Patients with early SGC had impaired survival compared to GC patients with the same UICC stage, calling for consideration of localization in staging systems in the future.

## Figures and Tables

**Figure 1 biomedicines-12-02136-f001:**
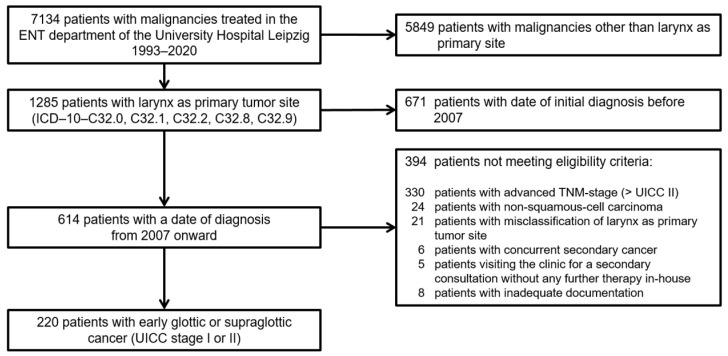
CONSORT diagram showing the selection process from registry data selecting 220 patients with early glottic and supraglottic cancer for analyses.

**Figure 2 biomedicines-12-02136-f002:**
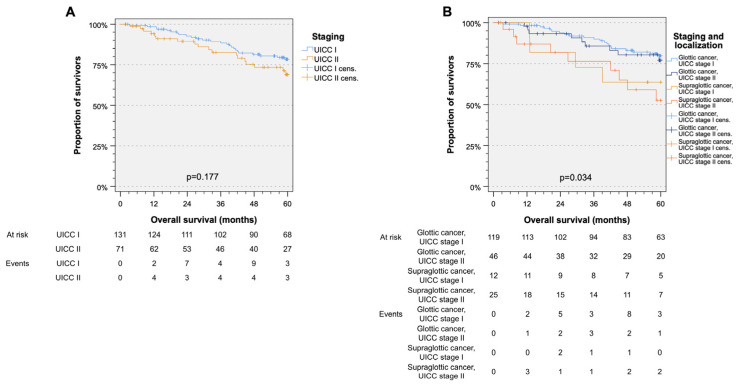
Kaplan-Meier cumulative survival plots addressing overall survival (OS) depending on UICC stage and localization. OS showed no significant difference between UICC stage I and II (**A**). Further differentiation between UICC stage and localization revealed a significantly different OS in patients with glottic and supraglottic cancer without systematic impact of UICC stage (**B**).

**Figure 3 biomedicines-12-02136-f003:**
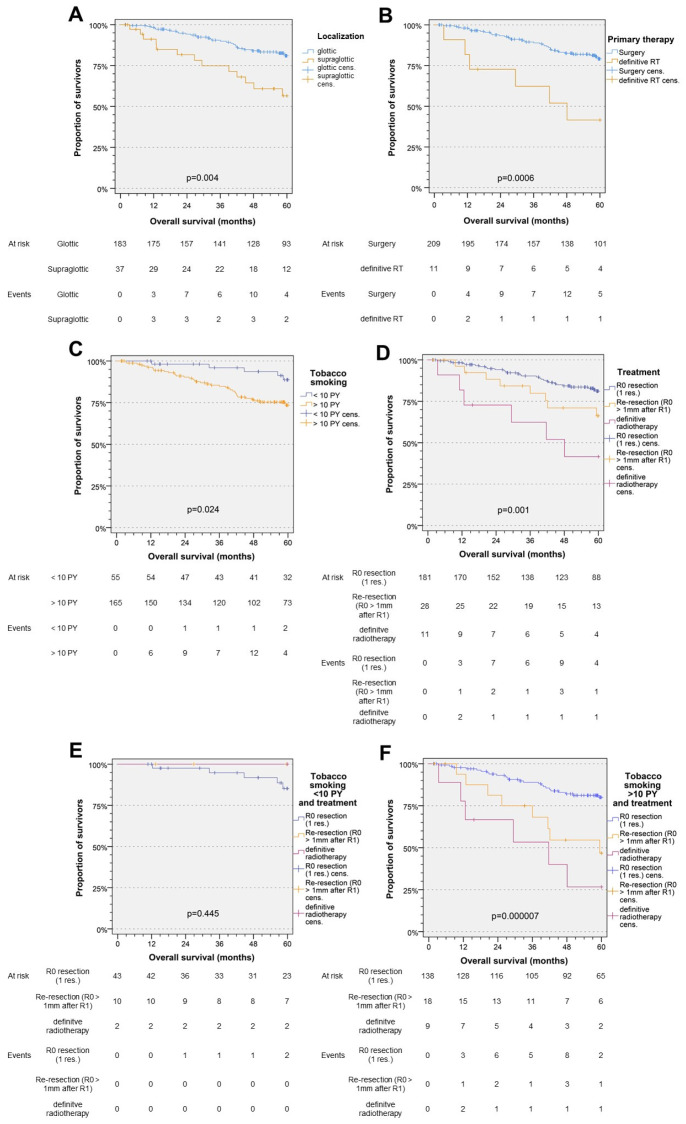
Kaplan-Meier curves addressing overall survival (OS) depending on localization (**A**) and primary therapy (**B**). The effect of tobacco smoking (**C**) and treatment with R0-resection, re-resection, or no surgery (**D**) on OS shows significant worse OS in heavy smokers (**E**,**F**).

**Figure 4 biomedicines-12-02136-f004:**
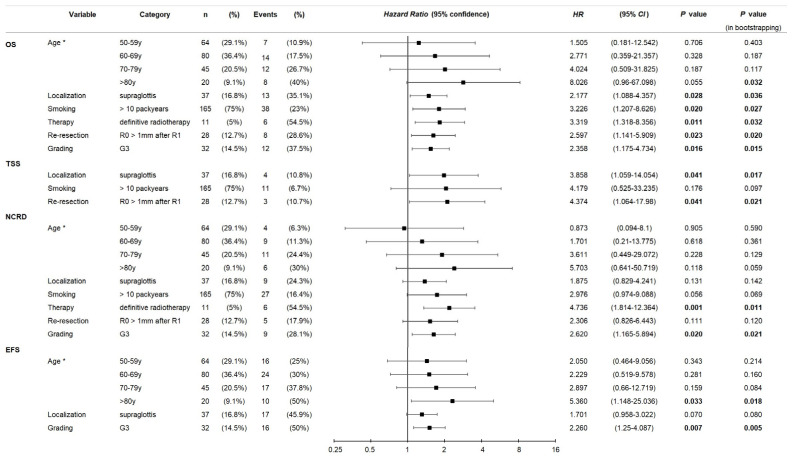
Forest plots from multivariate *Cox* proportional hazard regression models for outcome of early larynx cancer built via the automatic stepwise forward likelihood ratio method for covariate extraction generally demonstrate a higher hazard ratio for supraglottic cancer (ICD10-C32.1) according to overall survival (OS), tumor-specific survival (TSS), non-cancer-related death (NCRD), and event-free survival (EFS). Reference groups: age * ≤ 49 years; nonsmoking or smoking ≤ 10 pack years; localization glottis; grading G1/G2; treatment: surgery; internal validation of multivariate models through bootstrapping applying 1000 iterations; *p* values from 2-sided tests, significant *p* values in bold.

**Table 1 biomedicines-12-02136-t001:** The characteristics of the patients with early glottic and supraglottic cancer (UICC stage I and II) treated the University Hospital Leipzig since 2007 (*n* = 220). *p* values in bold indicate statistical significance.

						Localization			
			Total		Glottis		Supraglottis		*p* Value
		n	220	100.0%	183	83.2%	37	16.8%	
Patient Characteristics									
	Age [years]	<50	11	5.0%	10	5.5%	1	2.7%	0.667
		50–59	64	29.1%	50	27.3%	14	37.8%	
		60–69	80	36.4%	67	36.6%	13	35.1%	
		70–79	45	20.5%	38	20.8%	7	18.9%	
		>80	20	9.1%	18	9.8%	2	5.4%	
	Sex	Male	203	92.3%	173	94.5%	30	81.1%	**0.005**
		Female	17	7.7%	10	5.5%	7	18.9%	
Risk factors									
	Smoking	0	39	17.7%	37	20.2%	2	5.4%	**0.029**
	[pack years]	≤10	16	7.3%	14	7.7%	2	5.4%	
		>10	165	75%	132	72.1%	33	89.2%	
	Alcohol	0 g/d	50	22.7%	42	23.0%	8	21.6%	0.100
		0–30 g/d	95	43.2%	86	47.0%	9	24.3%	
		30–60 g/d	35	15.9%	26	14.2%	9	24.3%	
		>60 g/d	33	15.0%	26	14.2%	7	18.9%	
		Missing	7	3.2%	3	1.6%	4	10.8%	
Therapy									
	Surgery	No	11	5.0%	8	4.4%	3	8.1%	0.342
		Yes	209	95.0%	175	95.6%	34	91.9%	
	Neck Dissection	No	130	59.1%	122	66.7%	8	21.6%	**<0.001**
		Unilateral	46	20.9%	36	19.7%	10	27.0%	
		Bilateral	44	20.0%	25	13.7%	19	51.4%	
	Radiotherapy	No	198	90.0%	169	92.3%	29	78.4%	**0.006**
		Yes	22	10.0%	14	7.7%	8	21.6%	
	Chemotherapy	No	219	99.5%	183	100.0%	36	97.3%	**0.026**
		Yes	1	0.5%	0	0.0%	1	2.7%	
	Tracheostomy	No	166	75.5%	144	78.7%	22	59.5%	**0.013**
		Yes	54	24.5%	39	21.3%	15	40.5%	
Tumor Characteristics									
	TNM category	Tis	18	8.2%	18	9.8%	0	0.0%	**<0.001**
		T1a	101	45.9%	101	55.2%	0	0.0%	
		T1b	18	8.2%	18	9.8%	0	0.0%	
		T1	12	5.5%	0	0.0%	12	32.4%	
		T2	71	32.3%	46	25.1%	25	67.6%	
		N0	220	100.0%	183	100.0%	37	100.0%	
		M0	220	100.0%	183	100.0%	37	100.0%	
	UICC	Stage 0	18	8.2%	18	9.8%	0	0.0%	**<0.001**
		Stage I	131	59.5%	119	65.0%	12	32.4%	
		Stage II	71	32.3%	46	25.1%	25	67.6%	
	Grading	SIN III	18	8.2%	18	9.8%	0	0.0%	**0.009**
		G1	24	10.9%	21	11.5%	3	8.1%	
		G2	146	66.4%	123	67.2%	23	62.2%	
		G3	32	14.5%	21	11.5%	11	29.7%	

## Data Availability

The original contributions presented in the study are included in the article, further inquiries can be directed to the corresponding author.

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
