# Peer review of "Outcome Disparities in Patients with Early-Stage Laryngeal Cancer Depending on Localization, Tobacco Consumption, and Treatment Modality"

_biomedicines, 2024, doi:10.3390/biomedicines12092136_

Round 1
Reviewer 1 Report
Comments and Suggestions for Authors
Dear authors,
Thanks for your good work
A few comments to be noted
1- Title: please modify it to "Factors affecting outcome disparities in patients with early-stage laryngeal cancer"
2- Why were the patients who were upstaged pathologically excluded? Please explain
3- Also, why are the patients with Tis included? Their outcomes are usually excellent whatever the circumstances
4- for the patients who underwent R1 resection, why did not they undergo total laryngectomy?
5- Please specify the age <50/ Are they all 40-50 or the cohort included patients of younger ages
6- The discussion -despite being rich- needs to be relatively shorter
7- Please add the study limitations to the end of the discussion.
Author Response
Dear authors,
Thanks for your good work
Response to Reviewer’s comment: Many thanks for the comment!
A few comments to be noted
1- Title: please modify it to "Factors affecting outcome disparities in patients with early-stage laryngeal cancer"
Response to Reviewer’s comment: Many thanks for the comment! Of course, we can easily replace the title according to you recommendation. However, from our point of view, the main findings were the outcome disparities between T1 glottic and T2 supraglottic as well as T2 glottic and T2 supraglottic larynx cancers and the strong impact of tobacco smoking history (> 10 pack years vs. others) and treatment (resection vs, irradiation). As the outcome (survival) of patients and not the disparities are affected, and ‘factor’ could be deemed to be rather too unspecific (and not a MESH term searched by head and neck cancer oncologists) we would prefer to name the most important factors.
2- Why were the patients who were upstaged pathologically excluded? Please explain
Response to Reviewer’s comment: Many thanks for the question. By trying to present data about outcome disparities in early larynx cancer, and laryngeal squamous cell carcinoma (LSCC) in particular, we had to exclude all those patients with locally advanced (LA) LSCC. Otherwise we would have analyzed a biased mixture of stages, and the question would have been why we did not analyze all LSCC including all advanced LSCC independent from the timing within the clinical/diagnostic workflow. We clearly wanted to report outcome data solely of pathological confirmed early LSCC.
3- Also, why are the patients with Tis included? Their outcomes are usually excellent whatever the circumstances
Response to Reviewer’s comment: Many thanks for both the question and the comment! Of course, patients with carcinoma in situ at time of first diagnostic biopsy that get the precancerous lesion completely resected have excellent outcome and usually only require clinical control via regular microlaryngoscopy. However, some laryngeal Tis progress to LSCC that may or may not have impaired outcome. It might be in some respect difficult to report about early LSCC without mentioning Tis, especially in the light that those Tis progressing to T1 (most often glottic) LSCC. In such cases, the earlier information becomes overwritten by documentation of the first-time diagnosis of invasive cancer. Therefore, reporting the fraction of Tis not progressing to invasive LSCC provides welcome information about the frequency of such patients.
4- for the patients who underwent R1 resection, why did not they undergo total laryngectomy?
Response to Reviewer’s comment: Many thanks for the question. According to our national guidelines, total laryngectomy is overtreatment for early LSCC. Open partial resection could have been an alternative to irradiation but mostly is not accepted by LSCC patients after consultation of both specialists, ENT surgeons and radiation oncologists, that in such cases is always recommended by our multidisciplinary tumorboard (MDTB).
5- Please specify the age <50/ Are they all 40-50 or the cohort included patients of younger ages
Response to Reviewer’s comment: Many thanks for the comment! We added the range of age (44.2-87.3 years) in the Results-section, line 148.
6- The discussion -despite being rich- needs to be relatively shorter
Response to Reviewer’s comment: Many thanks for the comment. The advantage of open access publishing of online-only available content is the possibility to provide more information than the truncated discussion sections of papers published in printed form and therefore unavoidable limitations in the number of words allowed. The Discussion section, when considered being rich regarding the provided information and not redundant in arguments provided or papers cited, does not necessarily need to be shortened. As reviewer #2 did not ask for deletion of some aspects discussed or shortening the paper and Discussion in particular, we would like to keep the Discussion mostly as it is but delete some unnecessary words in lines 328 to 338. However, we follow your recommendation to provide more information about potential bias and other limitations at the end of the Discussion section. Of course, this wil rather contribute to a slightly longer Discussion section.
7- Please add the study limitations to the end of the discussion.
Response to Reviewer’s comment: Many thanks for the comment! We revised the Discussion section and provided more information about potential biases that might be present in our research and other limitations that may have affected the outcome of our investigation at the end of the Discussion section.

Reviewer 2 Report
Comments and Suggestions for Authors
I would like to thank the authors for their valuable efforts in this study. In the present study, the authors aimed to present their case history in the treatment of early stage laryngeal cancer, specifically evaluating patients with supraglottic and glottic localization tumors. They analyzed the results by comparing various treatment modalities and risk factors.
Among the data that emerged, it was found that tobacco users (over 10 PY) had a lower overall survival rate among patients treated with RHT. These results seem to suggest that smoking patients may benefit more from surgical treatment than radiotherapy.
These result seem to be very interesting.
I only have one note to make about self-citation. The authors cite two previously published studies by themselves (lines 64 and 380). If it is not necessary to explain the result obtained, self-citation should be strongly avoided. Moreover, the subject of the study is early stage (and not advanced) laryngeal cancer. For this reason, I would recommend removing or reducing the part from line 55 to line 66, as it is off-topic. Self-citation should be avoided in the manuscript. However, the one in line 380 is appropriate in the discussion of the results.
Author Response
I would like to thank the authors for their valuable efforts in this study. In the present study, the authors aimed to present their case history in the treatment of early stage laryngeal cancer, specifically evaluating patients with supraglottic and glottic localization tumors. They analyzed the results by comparing various treatment modalities and risk factors.
Among the data that emerged, it was found that tobacco users (over 10 PY) had a lower overall survival rate among patients treated with RHT. These results seem to suggest that smoking patients may benefit more from surgical treatment than radiotherapy.
These result seem to be very interesting.
I only have one note to make about self-citation. The authors cite two previously published studies by themselves (lines 64 and 380). If it is not necessary to explain the result obtained, self-citation should be strongly avoided. Moreover, the subject of the study is early stage (and not advanced) laryngeal cancer. For this reason, I would recommend removing or reducing the part from line 55 to line 66, as it is off-topic. Self-citation should be avoided in the manuscript. However, the one in line 380 is appropriate in the discussion of the results.
Response to Reviewer’s comment: Many thanks for the comment! According to your advice, we should revise the Introduction section, especially lines 55 to 66, as you deem the explanation that other treatment options have to be judged as overtreatment is off-topic. Indeed, we have most often published papers about locoregional advanced (LA) head and neck squamous cell carcinoma (SCC) including larynx and hypopharynx cancer (LHSCC) patients including a recent published paper about outcome in LHSCC treated with total laryngectomy (TL) followed by postoperative radiotherapy (PORT) or radio-chemotherapy (PORCT) or neoadjuvant (induction) chemotherapy or concurrent chemo-radiotherapy (CRT). As biomedicines is a journal addressing a broad readership potentially not familiar with head and neck cancer and treatment options for head and neck squamous cell carcinoma available and early vs. advanced stages in particular, we think it is necessary to provide this background information. However, deleting the sentence including reference [9] is no problem for us, and we followed your recommendation in this regard.
